# TOR Signaling Tightly Regulated Vegetative Growth, Conidiation, Oxidative Stress Tolerance and Entomopathogenicity in the Fungus *Beauveria bassiana*

**DOI:** 10.3390/microorganisms11092129

**Published:** 2023-08-22

**Authors:** Lai-Hong So, Jiraporn Jirakkakul, Lakha Salaipeth, Wachiraporn Toopaang, Alongkorn Amnuaykanjanasin

**Affiliations:** 1National Center for Genetic Engineering and Biotechnology (BIOTEC), National Science and Technology Development Agency (NSTDA), 113 Thailand Science Park, Paholyothin Rd., Tambon Khlong Nueng, Amphoe Khlong Luang, Pathum Thani 12120, Thailandwachiraporn.too@biotec.or.th (W.T.); 2School of Bioresources and Technology, King Mongkut’s University of Technology Thonburi, Bangkok 10140, Thailand; jiraporn.jir@mail.kmutt.ac.th (J.J.); lakha.sal@kmutt.ac.th (L.S.)

**Keywords:** *Beauveria bassiana*, CRISPR/Cas9, culture degeneration, Tet-On, TOR signaling

## Abstract

*Beauveria bassiana* degenerates after repeated subcultures, demonstrating declined conidiation and insect virulence. The target of rapamycin (TOR) kinase conserved among eukaryotes is the master regulator of cellular physiology and is likely involved in culture degeneration. Indeed, the levels of TOR-associated proteins increase over successive subcultures. Here, CRISPR/Cas9 locus engineering introduced the inducible Tet-On promoter upstream of the TOR kinase 2 gene *tor2* in *B*. *bassiana*. The mutant *P*^Tet-On^ *tor2* ‘T41’ was verified for the Tet-On integration via PCR analyses and provided a model for evaluating the fungal phenotypes according to the *tor2* expression levels, induced by doxycycline (Dox) concentrations. At 0 µg·mL^−1^ of Dox, T41 had 68% of the wild type’s (WT) *tor2* expression level, hampered radial growth and relatively lower levels of oxidative stress tolerance, conidiation and virulence against *Spodoptera exigua*, compared to those under the presence of Dox. A low dose of Dox at 0.1–1 µg·mL^−1^ induced *tor2* upregulation in T41 by up to 91% compared to 0 µg·mL^−1^ of Dox, resulting in significant increases in radial growth by 8–10% and conidiation by 8–27%. At 20 µg·mL^−1^ of Dox, which is 132% higher than T41’s *tor2* expression level at 0 µg·mL^−1^ of Dox, T41 showed an increased oxidative stress tolerance and a decrease in growth inhibition under iron replete by 62%, but its conidiation significantly dropped by 47% compared to 0 µg·mL^−1^ of Dox. T41 at 20 µg·mL^−1^ of Dox had a strikingly increased virulence (1.2 day lower LT_50_) against *S*. *exigua*. The results reflect the crucial roles of TOR kinase in the vegetative growth, conidiation, pathogenicity and oxidative stress tolerance in *B*. *bassiana*. Since TOR upregulation is correlated with culture degeneration in multiple subcultures, our data suggest that TOR signaling at relatively low levels plays an important role in growth and development, but at moderate to high levels could contribute to some degenerated phenotypes, e.g., those found in successive subcultures.

## 1. Introduction

The culture degeneration or attenuation of the entomopathogenic fungus *Beauveria bassiana* has long been reported to be parallel to its widespread application in integrated pest management (IPM). Several fungi degenerate after successive subculturing in vitro, such as on potato dextrose agar (PDA) and Sabouraud dextrose agar (SDA). The degenerated cultures show a reduction or loss in virulence [1,2], which is partially attributed to the reduced secretion of host-cuticle-degrading protease and a reduction in conidiation [3,4]. Entomopathogenic fungi with inconsistent conidial yield or virulence become unattractive to agricultural practitioners [5].

The physiological changes in degenerated *Beauveria bassiana* cultures were recently unraveled via proteomics analysis. Late in vitro subcultures had increased levels of proteins, which were associated with oxidative stress response, autophagy, amino acid homeostasis and apoptosis, whilst decreased levels of proteins were associated with DNA repair, ribosome biogenesis, energy metabolism and virulence [6]. The finding that two effectors of target of rapamycin (TOR) signaling were elevated in late subcultures suggested the contribution of TOR signaling components in *B. bassiana* degeneration.

TOR signaling is the master regulator of growth and is highly conserved in eukaryotes [7]. It is mediated by two functionally distinct complexes, TOR complex 1 (TORC1) and TORC2. The better understood TORC1, which is rapamycin sensitive, responds to nutrient availability and promotes growth by stimulating nutrient uptake and the synthesis of proteins, lipids and nucleotides [8]. TORC2, which is marginally rapamycin sensitive, coordinates with its major downstream effector Ypk1 and serves as the major regulator of plasma membrane lipid homeostasis [9]. Intriguingly, numerous studies have suggested that TORC1 inhibition effectively extended the lifespan across different eukaryotes such as yeast, *Caenorhabditis elegans, Drosophila melanogaster* and mice [10,11,12,13]. Nonetheless, few reports establish a link between TOR inhibition and culture degeneration, analogous to the in vitro lifespan, in filamentous fungi.

*Beauveria bassiana* BCC 2660 has a single copy of the TOR kinase gene, *tor2*, which serves as a common core component of TORC1 and TORC2, similarly to most eukaryotes. In this study, a *B. bassiana P*^Tet-On^ *tor2* mutant was used to investigate exogenous *tor2* expression induction [14], which helped to document a correlation between *tor2* expression levels and the affected phenotypes as a result of culture degeneration. We aim to determine the contribution of TOR kinase in growth and development in the fungus, and to determine if the relevant contributions are dose dependent. We demonstrate the significance of *tor2* in mycelial growth, conidiation, virulence and oxidative stress tolerance in this fungus. 

## 2. Materials and Methods

### 2.1. Fungal/Bacterial Strains and Culture Conditions

*Beauveria bassiana* BCC 2660 was acquired from BIOTEC Culture Collection (BCC), Thailand and was maintained on potato dextrose agar (PDA; Difco, Franklin Lakes, NJ, USA) at 25–28 °C. To ensure the highest virulence, the fungus was first passaged through *Spodoptera exigua* following Jirakkakul et al.’s (2018) [6] protocol. In subculturing, conidia from 7-day-old cultures were harvested. Conidial suspension containing 1 × 10^7^ conidia was spread on half-strength PDA and incubated for 7 days at 25–28 °C. Wild-type (WT) strain of *B. bassiana* BCC 2660 was subcultured from the 2nd to 20th subcultures for the examination of culture degeneration. For isolation of genomic DNA or total RNA, *B. bassiana* was shaken in potato dextrose broth (PDB) at 150 rpm at 25–28 °C for 4 days. The Tet-On inducer, doxycycline (Dox), at 0, 0.1, 0.5, 1 or 20 µg·mL^−1^ was also supplemented in culture medium for total RNA extraction. 

*Escherichia coli* DH5α was routinely used in molecular cloning. Culture media were supplemented with ampicillin for the recombinant DNA cloning of pCB1534 (Fungal Genetic Stock Center, Kansas, MO, USA; [15]), pFC332 and pFC334 [16]. 

### 2.2. Construction of CRISPR/Cas9 tor2-Targeted Vector and Tet-On-Carrying Donor Vector

CRISPR/Cas9-directed double-stranded break (DSB) and homology-directed repair were adopted to introduce the Tet-On promoter just upstream of the *tor2* coding sequence. The endogenous *tor2* promoter remained since it was shared between *tor2* and an adjacent gene, encoding a hypothetical protein. The *tor2*-targeted CRISPR/Cas9 vector was cloned by following Nødvig et al.’s (2015) protocol and using the plasmids pFC332 and pFC334 [16]. Two fragments were amplified for the construction of *tor2*-targeted sgRNA cassette. The first fragment was amplified with primers CSN389 (5′-GGGTTTAAUGCGTAAGCTCCCTAATTGGC-3′) and tor2sgRNA-r (5′-AGCTT ACUCGTTTCGTCCTCACGGACTCATCAGGTCACTCGGTGATGTCTGCTCAGCG-3′). The second fragment was amplified with primers tor2sgRNA-f (5′-AGTAAGCUCGTCGTCACTGAATTGGGAAACCGGTTTTAGAGCTAGAAATAGCAAGTTAAA-3′) and CSN390 (5′-GGTCTTAAUGAGCCAAGAGCGGATTCCTC-3′). The two fragments integrated the *tor2*-specific protospacer sequence (5′-GTCACTGAATTGGGAAACCGGGG-3′; PAM site is CCG; Figure 1A), and were then cloned into the backbone vector pFC332 at the restriction site PacI/Nt.BbvCI via USER cloning (New England Biolabs, Ipswich, MA, USA), resulting in the recombinant plasmid pFC-TOR2.

The donor DNA vector contains the 2581 bp Tet-On sequence, flanked by a 991 bp 5′-homology arm (5′-HA) and a 900 bp 3′-homology arm (3′-HA) that have sequences complementary to the *tor2* promoter sequences and the 5′ end of the *tor2* coding sequence, respectively. For the construction of the donor DNA vector, the three DNA fragments were cloned into the vector pCB1534 via restriction cloning. The primers 5F-Bsi (5′-GATCGTACGCTGATGCTTCTCGAATCTGC-3′) and 5R-Nco (5′-GTACCATGG AAGCAAGCTTGCCCGCGCAA-3′) were used to amplify the 5′-HA from the genomic DNA of *Beauveria bassiana* BCC 2660. The primers TetF-Nco (5′-GAACCATGGAA GCTTCCCTCGGCTGGTCTGT-3′) and TetR-Age (5′-GACACCGGTGTTTAAAC GGTGATGTCTGCT-3′) were used to amplify the Tet-On sequence from pFW22.1, a plasmid vector that carries the Tet-On inducible gene expression system. Lastly, the primers 3F-Age (5′-GATACCGGTAACTGGCGAGGACAGTGTGCA-3′) and 3R-Avr (5′-GCGCCTAGGTTCTTGTTTCCAGGTGTAATGC-3′) were used to amplify 3′-HA from the *B. bassiana* genomic DNA. After PCR amplification, the 5′-HA fragment was introduced to pCB1534 via double digestion of both pCB1534 and 5′-HA by *Bsi*WI and *Nco*I, followed by DNA ligation. This step generated the recombinant vector pCB-5HA. Next, the Tet-On sequence was introduced via double digestion of both pCB-5HA and the Tet-On fragment by *Nco*I and *Age*I, followed by DNA ligation. The cloning generated pCB-5HA-Tet. Lastly, 3′-HA was introduced via double digestion of both pCB-5HA-Tet and 3′-HA by *Age*I and *Avr*II and DNA ligation, hence generating pCB-5T3 (Figure 1A). Both the CRISPR/Cas9 vector pFC-TOR2 and the donor vector pCB-5T3 were verified via DNA sequencing.

### 2.3. Transformation of Beauveria bassiana Protoplasts with the CRISPR/Cas9 Vector and Donor Vectors

The CRISPR/Cas9 vector and the donor vector were co-introduced to *Beauveria bassiana* via PEG-mediated protoplast transformation with modifications from Srisuksam et al.’s (2015) protocol [17]. Conidial suspension containing 1 × 10^8^ conidia from 5-day culture was inoculated to 50 mL of PDB. The culture was shaken at 150 rpm at 25 °C for 16 h, and the young mycelia were harvested via centrifugation at 1500× *g*. For protoplast preparation, cell-well-degrading enzyme mix VinoTaste Pro (Novozymes, Frederiksberg, Denmark) was added at a final concentration of 35 mg·mL^−1^ in 0.6 M MgSO_4_. The mix was incubated at 28 °C at 90 rpm until most of the mycelia were digested to protoplasts. Equal volumes of 0.6 M sorbitol and 100 mM Tris buffer (pH 7.0) were added, followed by centrifugation at 1500× *g* to collect the protoplast pellet. Protoplasts were then resuspended in 500 µL of 1 M sorbitol, 10 mM Tris buffer (pH 7.5) and 10 mM CaCl_2_. Then, the protoplasts were added to a transformation mixture comprising 50 µL of transformation solution (40% PEG (50%, *v*/*v*), 0.6 M KCl (15%, *v*/*v*), 50 mM Tris buffer (pH 8.0) (5%, *v*/*v*), 50 mM CaCl_2_ (5%, *v*/*v*), water (25%, *v*/*v*)), CRISPR/Cas9 vector and donor vector (20 µg each). After incubation on ice for 30 min, an additional 1 mL of transformation solution was added to the mix and it was left to incubate in RT for 45 min. Transformation mix was subsequently transferred to 15 mL of molten PDA supplemented with 0.8 M sucrose and 250 µg·mL^−1^ hygromycin B (hyg) before pouring into a Petri dish. The culture was incubated at 28 °C until emergence of fungal colonies. 

### 2.4. Beauveria bassiana P^Tet-On^ tor2 Mutant Selection and Verification

Fungal colonies from transformation culture were isolated and transferred to new selection media (PDA with 250 µg mL^−1^ of hyg). Hyg selection was repeated for five generations. Afterwards, colonies were propagated on half PDA and then in PDB for genomic DNA isolation.

Tet-On introduction in transformants was verified by two rounds of PCR. The first PCR verified the presence of Tet-On promoter in transformants, using *Tet* primer pair HindIII_Tet_F (5′-AAGCTTCCCTCGGCTGGTCTGTCTTAC-3′) and Tet_plus_R (5′-ATCCAGATTGCACACTGTCCGTTTAAACGGTGATGTCTGCTC-3′). The second PCR verified Tet-On insertion downstream of *tor2* promoter. The primer pair TorTet-seqF (5′-GCAATACCAGCCACACCTGCA-3′) and TorTet-seqR (5′-GCGAGTAACT AGCCAGGTCAG-3′) amplified partial *Tet* sequence and *tor2* coding sequence. Transformants showing correct DNA band sizes in both PCR were subjected to Sanger DNA sequencing.

### 2.5. Gene Expression Analyses of tor2 in P^Tet-On^ tor2 Mutant under Different Dox Inductions

*P*^Tet-On^ *tor2* mutants induced with 0, 0.1, 0.5, 1 and 20 µg·mL^−1^ of Dox were determined for their *tor2* expression levels using qPCR. Total RNA was extracted with Ambion TRIzol Reagent (Thermo Fisher Scientific, Waltham, MA, USA). Traces of DNA were removed using Ambion DNase I (Thermo Fisher Scientific). First-strand cDNA was then synthesized using RevertAid First Strand cDNA Synthesis Kit (Thermo Fisher Scientific). All procedures were conducted according to the manufacturer’s protocol. First-strand cDNA from samples were verified via PCR using *tor2*-specific primer pair tor2-f (5′-GCAATGTCAGCGTCCAGAAT-3′) and tor2-r (5′-AAAAGTACCCGGAGCGT GTA-3′), as well as the housekeeping gene, M1PD (mannitol 1-phosphate dehydrogenase). The primer pairs amplified intron-inclusive coding sequence in genome so as to distinguish PCR product bands between *tor2* cDNA and control (genomic DNA). qPCR was performed using SYBR Green Master Mix (Bio-Rad, Hercules, CA, USA). Thermal cycle was as follows: 95 °C for 10 min; 39 cycles of 95 °C for 15 s; 55 °C for 30 s; and 72 °C for 30 s. M1PD was used as internal control. Expression fold change of *P*^Tet-On^ *tor2* mutant treatments relative to WT was normalized and calculated using the 2^–ΔΔCt^ method [18]. Experiments were repeated twice.

### 2.6. Radial Growth Assays and Oxidative Stress Tolerance Analysis of P^Tet-On^ tor2 Mutant

Radial growth assay for *P*^Tet-On^ *tor2* mutant was conducted to determine optimal Dox concentration and to assess *tor2*-associated radial growth change. Conidial drop prepared as above was inoculated to PDA supplemented with 0, 0.1, 0.5, 1, 5, 50, 100, 200 and 500 µg·mL^−1^ of Dox. WT was used as comparative control. Colony diameter was measured daily. Each treatment consisted of 4 replicates, and the experiment was repeated twice.

Tolerance of *P*^Tet-On^ *tor2* mutant against oxidative stress under different *tor2* inductions was assessed. Mutant was cultured with 0 or 20 µg·mL^−1^ of Dox and was compared to WT. Ferrous ion (Fe^2+^) was introduced as FeSO_4_ at 0 or 10 µM to induce oxidative stress. Ten-microliter drops of conidial suspension at 1 × 10^7^ conidia·mL^−1^ were cultured on minimal media agar (0.02% (*w*/*v*) dextrose, 0.51% (*w*/*v*) (NH_4_)_2_SO_4_, 0.17% (*w*/*v*) yeast nitrogen base (YNB) without amino acids) and incubated at 25–28 °C for 12 days with daily examination and measurement of colony diameter. Each datum was averaged from two perpendicular measurements. Each treatment consisted of 3 replicate plates with 2 colonies per replicate, and the experiment was repeated twice.

### 2.7. Conidiation Analyses of P^Tet-On^ tor2 Mutant and Beauveria bassiana WT Culture Degeneration

Subcultures of *Beauveria bassiana* WT or *P*^Tet-On^ *tor2* mutant were cultured on PDA for 7 days. Mutant *tor2* was induced by 0, 0.1, 0.5, 1 or 20 µg·mL^−1^ of Dox. For successive subcultures of WT, subcultures S2, S4, S8, S12, S16 and S20 were analyzed. Two hundred microliters of conidial suspension at 5 × 10^7^ conidia·mL−1 was spread on PDA plate and incubated at 28 °C for 7 days. Conidia were then harvested by rinsing with 10 mL of 0.1% Tween 20 and counted. Each treatment consisted of 3 replicates, and the experiment was repeated twice.

### 2.8. Insect Bioassay of Beauveria bassiana P^Tet-On^ tor2 Mutant and WT 

Fungal culture and *tor2* induction were prepared similarly as above. For *B. bassiana* WT, S2, S8, S12 and S20 were analyzed. Conidial suspension was prepared with saline (0.85% NaCl) and was adjusted to 1 × 10^5^ conidia·mL^−1^. Conidia were introduced to the larvae of *Spodoptera exigua* via intrahemocoelic injection, adapted from Toopaang et al.’s (2017) protocol [19]. Fourth-instar beet armyworm (*S*. *exigua*) larvae were obtained from the Viral Technology for Biocontrol Research Team (BIOTEC, Pathum Thani, Thailand). Three microliters of the conidial suspension was injected to the insect larvae with a specialized 33-gauge microliter syringe set (Hamilton, NV, USA). Injected larvae were then transferred to individual wells of a 24-well plate coated with mung-bean-based semi-synthetic diet (in 1 L: (*w*/*v*) 13% mung bean, 1% brewer’s yeast, 0.3% ascorbic acid, 0.125% sorbic acid, 0.25% methyl paraben, 0.3% casein, 0.15% vitamin E, 0.05% choline chloride and 1.2% agar; (*v*/*v*) 0.3% vitamin stock and 0.08% formalin). Control larvae were injected with saline. Each treatment consisted of ten larvae. Cumulative insect mortality was recorded for 7 days on a daily basis. The experiment was repeated twice. Median lethal time (LT_50_) was determined using SPSS’s probit analysis.

### 2.9. Statistical Analyses

Experiments in this study were conducted either two or three times with three or four replicates for each treatment. Statistical significance was determined using one-way ANOVA. Univariate linear model was used, and Duncan’s test was used as the post hoc test. The significance letters represent the different homogenous subsets in Duncan’s post hoc test. LT_50_ in insect bioassay was calculated using Probit regression analysis. LT_50_ was determined by recording the days required for killing half of the beet armyworm population. All significance tests were completed using SPSS package version 23 (IBM, Chicago, IL USA).

## 3. Results

### 3.1. CRISPR/Cas9-Mediated Tet-On Was Introduced at the Upstream Site of tor2 Coding Sequence

Eighty transformants were initially recovered under hygromycin selection after PEG-mediated protoplast transformation. Four additional rounds of repeated selections using hygromycin screened out 71 colonies of ectopic integration, whereas the remaining 9 colonies were subject to PCR. In the first PCR, the transformant T41 displayed the correct Tet-On PCR product size of 2581 bp, and the transformant T54 also showed the expected product with less intensity (Figure 1B,C). In the second PCR, the transformants T32 and T41 showed the correct product size of 1461 bp (Figure 1D). The similarly sized products with noticeably less intensity were also detected in T38, T54, T65 and T74. Based on these two amplification results, the transformant T41 was selected as the *P*^Tet-On^ *tor2* model for the following experiments. The DNA sequencing confirmed the correct integration of the Tet-On cassette in the transformant.

### 3.2. Doxycycline Affected the Radial Growth in a Concentration-Dependent Manner

The radial growth determination revealed distinct patterns of radial growth changes between the *Beauveria bassiana* WT and the mutant *P*^Tet-On^ *tor2* in response to the Dox concentrations. Under the culture without Dox, the mutant had a markedly decreased growth compared to the WT (the radial growth values at 0 µg·mL^−1^ are shown; Figure 2A,B). When the mutant was supplemented with a low range of Dox concentrations at 0.1–1 µg·mL^−1^, it showed a similar growth to the WT on the Dox-free culture (Figure 2A,B), suggesting the range of concentrations required for the vegetative (or hyphal) growth in this fungus. At a high range of concentrations of Dox at 100–500 µg·mL^−1^, the growth inhibition was more pronounced in both the strains (Figure 2A,B). The WT demonstrated 15–19, 28–33 and 37–43% inhibition in radial growth at 100, 200 and 500 µg·mL^−1^ (Figure 2A), respectively. The mutant T41 showed 17–22, 28–35 and 41–52% inhibition at the same three concentrations (Figure 2B). The data remarkably suggest that the high concentrations of Dox at 100, 200 and 500 µg·mL^−1^ led to the growth inhibition at approximately 20, 30 and 40%, respectively.

### 3.3. RT-qPCR Confirmed Dox-Dependent tor2 Expression Induction in T41

The *tor2* expression in the *P*^Tet-On^ *tor2* strain T41 displayed Dox-dependent upregulation (Figure 3) across 0.1 to 20 µg·mL^−1^ of Dox. The Dox-free T41 culture showed the lowest *tor2* expression at 0.68-fold relative to the WT, indicating a 32% downregulation from the WT’s expression level. By contrast, a low range of concentrations of Dox at 0.1 and 1 µg·mL^−1^ induced 17% and 30% upregulation relative to the WT, respectively. An amount of 20 µg·mL^−1^ of Dox resulted in a 58% upregulation of *tor2* expression.

### 3.4. tor2 Contributed to Fungal Oxidative Stress Tolerance

The mutant T41 exhibited different degrees of oxidative stress tolerance depending on the *tor2* expression levels. Without the exogenous supplementation of ferrous ion, the WT, the T41 at 0 µg·mL^−1^ of Dox and the T41 at 20 µg·mL^−1^ of Dox did not show apparent morphological differences (Figure 4A). On the contrary, under iron replete, when 10 µM of Fe was introduced, the WT displayed considerable growth inhibition with a 15.2% decrease in radial growth on day 12, whereas the T41 at 20 µg·mL^−1^ of Dox was inhibited at a significantly lower degree (4.8%) of inhibition at the same incubation time (Figure 4B). In the absence of Dox, T41 showed the indistinguishable growth inhibition to that of the WT on day 12 under the iron replete. 

When overviewing the growth inhibition among treatments across day 3 to day 12, it can be seen that all the treatments gradually reduced the growth inhibition over the incubation time. The WT’s growth inhibition decreased from 30 to 21, 19 and 15% on days 3, 6, 9 and 12, respectively. The mutant T41 in the absence of Dox and at 20 µg·mL^−1^ of Dox also had a gradual decrease in the growth inhibition from 22 to 13% and from 15 to 5% on days 3 and 12, respectively. Collectively, the results reflect that *Beauveria bassiana* growth was inhibited by oxidative stress induced by ferrous ion introduction, but at the later stage, the subsequent oxidative stress response alleviated the inhibition in this fungus. The TOR kinase gene is likely involved in oxidative stress tolerance since *tor2* upregulation resulted in drastically reduced growth inhibition. The moderate expression of *tor2* induced by 20 µg·mL^−1^ of Dox could promote the oxidative stress tolerance in this fungus.

### 3.5. tor2 Was Associated with Fungal Conidiation

The *P*^Tet-On^ *tor2* mutant T41 exhibited distinct conidiation patterns dependent on Dox concentrations. Across all the T41 treatments, T41 at 0 µg·mL^−1^ of Dox had the lowest conidiation at 3.40 × 10^9^ conidia per Petri dish (Figure 5A). The Dox supplementation resulted in two distinct conidiation patterns. At a low range of Dox concentrations, 0.1/0.5/1 µg·mL^−1^, T41 conidiation gradually increased to 4.31 × 10^9^ conidia per Petri dish, marking a 27% increase compared to the T41 at 0 µg·mL^−1^ of Dox. However, at 20 µg·mL^−1^ of Dox, conidiation drastically dropped to 1.82 × 10^9^, representing a 47% decrease from that of the T41 at 0 µg·mL^−1^ of Dox.

The conidiation pattern of the in vitro passages demonstrated the culture degeneration of the *Beauveria bassiana* WT, as observed in the previous report [6]. Successive subcultures of the WT gradually reduced conidiation from the 2nd (S2) subculture at 13.8 × 10^9^ conidia per Petri dish to the 20th (S20) subculture at 2.8 × 10^9^ conidia per Petri dish, suggesting a notable decrease of 51% (Figure 5B). The conidiation of the late subcultures, S16 and S20, was comparable to the conidiation of T41 at 20 µg·mL^−1^ of Dox. The results together reflect that *tor2* was involved in *B. bassiana* conidiation. A mild level of *tor2* expression, induced by 0.1–1 µg·mL^−1^ of Dox, enhanced conidiation, but a higher *tor2* expression level at 20 µg·mL^−1^ of Dox intensely suppressed conidiation in this fungus.

### 3.6. tor2 Was Associated with Virulence against S. exigua

The virulence of the *P*^Tet-On^ *tor2* mutant T41 against the *S. exigua* larvae depended on the Dox-induced *tor2* expression levels. Among the different Dox treatments in T41, a low range concentration of Dox at 0, 0.1 and 1 µg·mL^−1^ showed similar mortality percentages to those of the WT (Figure 6A). The LT_50_ of these four treatments were 4.28, 3.65, 3.74 and 4.19, respectively (Table 1). In contrast, T41 at a moderate concentration of Dox at 20 µg·mL^−1^ had a strikingly increased mortality compared to the treatments at a low concentration range (0–1 µg·mL^−1^). Nonetheless, an even higher concentration of Dox at 50 µg·mL^−1^ led to lower mortalities in T41 than at 20 µg·mL^−1^, suggesting that an optimal *tor2* expression level, induced by a moderate concentration of Dox, supports the high insect mortality in *Beauveria bassiana*. A too-high level of *tor2* expression resulted in a lower insect mortality, as we also observed in the late subcultures.

T41 at 20 µg·mL^−1^ of Dox had the lowest LT_50_ at 3.08 days, which was 1.2 days (28.8 h) shorter than T41 in the absence of Dox (Table 1). 

For the culture degeneration in the *Beauveria bassiana* WT, successive subcultures of *B. bassiana* in vitro led to deteriorating virulence against *S. exigua*. From S2 to S20, the cumulative mortalities decreased over multiple passages in the culture (Figure 6B), consistently reflected in increased LT_50_ over the subcultures (Table 2). Compared to the LT_50_ of S2 at 3.55 days, the LT_50_ of S8, S12 and S20 were 3.63, 3.76 and 4.16 days, respectively, among which the LT_50_ of S20 was delayed from S2 by 0.61 days (14.6 h). The results overall reflect the contribution of *tor2* in *B. bassiana*’s virulence against insects.

## 4. Discussion

The morphological changes in degenerated fungal cultures include the attenuation of sporulation and impairment in sexuality, fruiting body formation and secondary metabolism [20]. It is important to understand the mechanisms contributing to culture degeneration, since the phenotypic stability of fungal strains in in vitro cultivation is industrially important in the manufacture of mycoinsecticides [21,22]. TOR signaling regulates fungal growth, energy metabolism, survival and the secondary metabolism in response to nutrient availability, growth factors and stresses; thus, TOR reasonably plays a potential role in maintaining the phenotypic stability in filamentous fungi. The deletion of the sole TOR kinase gene in filamentous fungi is lethal such as in *Fusarium fujikuroi*, *Aspergillus fumigatus*, *Trichoderma atroviridae*, *Verticilium dahliae* [23,24,25,26] and *Beauveria bassiana* BCC2660 (unpublished data). In this study, we selected the TOR kinase 2 gene ‘*tor2*’ to target both the TOR complexes that generally exist in most organisms, TORC1 and TORC2. The Tet-On introduction was precisely placed just upstream of the *tor2* coding sequence. The native *tor2* promoter remained at the locus since it is likely shared with an adjacent gene. The CRISPR/Cas9-mediated Tet-On promoter introduction enabled exogenous *tor2* expression induction via Dox. Here, our data from various characterizations of the *P*^Tet-On^ *tor2* mutant demonstrate the significance of the TOR kinase in *B. bassiana*’s mycelial growth, conidiation, virulence and oxidative stress tolerance. It is also interesting how this essential developmental program may contribute to the culture degeneration of *B. bassiana*.

Fungal conidiation, vegetative growth and oxidative stress tolerance predominately determine virulence against insect hosts. In nature, firstly, the fungal dispersal is principally carried out by means of ‘conidiation’ and subsequent release into the surrounding area. The dispersed *Beauveria bassiana* conidia attach to the host cuticle. Secondly, these cells germinate, grow ‘vegetatively’ and form the cuticle-penetrating appressorium for hemolymph access. The fungus then gradually infects and colonizes the host by exploiting the host’s nutrients for proliferation, releasing insecticidal compounds whilst evading the host’s immune response. The fungus later emerges from the host cadaver and releases conidia, where the infection cycle repeats [27]. Conidiation and hyphal growth therefore determine the success of conidial attachment to the host and pathogenesis. 

Thirdly, ‘oxidative stress tolerance’ plays an important role as the stress response when exposed to the host’s ‘oxidative burst’ upon microbial detection [28] or to adverse stresses from habitat niches [29]. TOR signaling coordinates these machineries in *Beauveria bassiana* through an intricate regulation network, as supported by knockdown studies. TORC1 inhibition via rapamycin often leads to a serious or complete loss of fungal sporulation. The antisense RNA knockdown of the TOR kinase gene in *B. bassiana* increased the fungus’s sensitivity to osmotic stress, heat stress, cell-wall-perturbing agents and oxidative stress [30]. The TOR-regulated transcription factor Msn2/4 negatively regulates the production of protease, lipase and oosporein in *B. bassiana* [31], which are important in facilitating host cuticle penetration and minimizing microbial competition in the host cadaver [31,32], while in the fungal plant pathogen *V. dahliae*, TORC1 inhibition via rapamycin significantly inhibits fungal mycelial growth and pathogenicity [26]. Similarly, in *Fusarium graminearum*, the TOR downstream component Tap42 interacts with different types of 2A phosphatases Pp2A, Sit4 and Ppg1, which regulate the cell wall integrity, mycelial growth, mycotoxin production and virulence [33]. TOR also regulates the secondary metabolism, including polyketide biosynthesis, in mycoparasitic fungus *T. atroviridae* [25], which is essential for its mycoparasitism. Therefore, TOR plays a crucial role in mediating fungal pathogenesis. 

In this study, the mutant *P*^Tet-On^ *tor2* ‘T41’ serves as a tool for evaluating the fungal phenotypes according to the TOR expression levels. Our results from these phenotypic characterizations support TOR’s significance in the growth, survival and pathogenesis of the fungus. *tor2’*s downregulation from the absence of Dox resulted in the hampering of *P*^Tet-On^ *tor2’*s radial growth, a lower oxidative stress tolerance and a decrease in conidiation and virulence against *S. exigua* compared to those under the presence of Dox. The upregulation of *tor2* induced by Dox resulted in different extents of improvement in these phenotypes depending on the Dox concentration. A low range of concentrations of Dox at 0.1–1 µg·mL^−1^ induced *tor2* upregulation by up to 30%, resulting in an increasing gradient of conidiation and radial growth. However, at a relatively higher Dox concentration at 20 µg·mL^−1^, the oxidative stress tolerance increased, but the conidiation significantly dropped. The conidiation decrease in T41 is akin to the deteriorating conidiation in the *Beauveria bassiana* WT after successive subcultures in vitro [6], supporting a notion of correlation of elevated TOR signaling in repetitive subcultures and degeneration in conidiation after a certain TOR expression threshold. However, the finding that virulence against *S. exigua* of T41 with 20 µg·mL^−1^ of Dox did not exhibit a similar drop to that found in conidiation suggests that the regulatory spectrum by the TOR signaling pathway may be different depending on the phenotypes. Alternatively, the TOR pathway may not be the sole factor for the declined insect virulence; instead, this may be caused by multiple factors that were not determined in our *P*^Tet-On^ *tor2* model.

Apart from its role in virulence, the TOR signaling pathway was reported to be associated with *Beauveria bassiana*’s culture degeneration [6]. TOR negatively regulates catabolic activities such as mRNA degradation, ubiquitin-dependent proteolysis and autophagy, which are potential stress-responsive cellular processes that may contribute to lifespan extension [34]. Temporally, TOR’s role in aging control may begin in adulthood. In *C. elegans*, TOR inhibition during development is growth inhibitory; yet, inhibition starting from the first day of adulthood lengthens its lifespan [11]. Our experimental data confirm TOR’s inhibitory effect on conidiation, radial growth and virulence when the TOR kinase gene is downregulated in early development. Further experiments on the effect of TOR inhibition on the degeneration of in vitro subcultures of *B. bassiana* could attest to TOR’s potential role in culture degeneration at ‘adulthood’, which we speculate to be a representation of the restriction of *B. bassiana* to a saprophytic mode of nutrition during cultivation on axenic media.

TOR’s contribution to culture degeneration could stem from the failure to select longevity-promoting traits in natural selection [35]. Under in vitro cultures, individuals with robust TOR activity are selected, which could drive cellular hyperfunction and pathogenicity [36]. This could explain some observations of the phenotypic and physiological changes in culture degeneration, including the change in the color of the conidia in *M. anisopliae*, the development of a sterile sector in *A. nidulans* as well as the altered energy metabolism in *Beauveria bassiana* [6,20,37]. Nonetheless, further studies need to be conducted to clarify the mechanisms of how degenerated cultures often restore to maximum virulence after the reintroduction to an insect host.

The culture degeneration of entomopathogenic fungi probably results from a combination of inter-connected factors [5], where strains, culture media and environmental cues could contribute to the degeneration in addition to TOR signaling. The reliance of stimuli from an insect host partially explains their ability to rejuvenate and restore to full pathogenicity potential after one or two passages through an insect host; yet, axenic culture media fail to select virulent offspring. The possible involvement of TOR in these processes raises a possible strategy of alleviating the fungal culture degeneration via TOR inhibition, as shown in other organisms. Indeed, our data demonstrate that a tight regulation of the TOR signaling pathway would be the key to rejuvenate these fungal cultures. Our study aims to shed light on the culture degeneration in entomopathogenic fungi and the linkage between TOR, aging and culture degeneration.

## Figures and Tables

**Figure 1 microorganisms-11-02129-f001:**
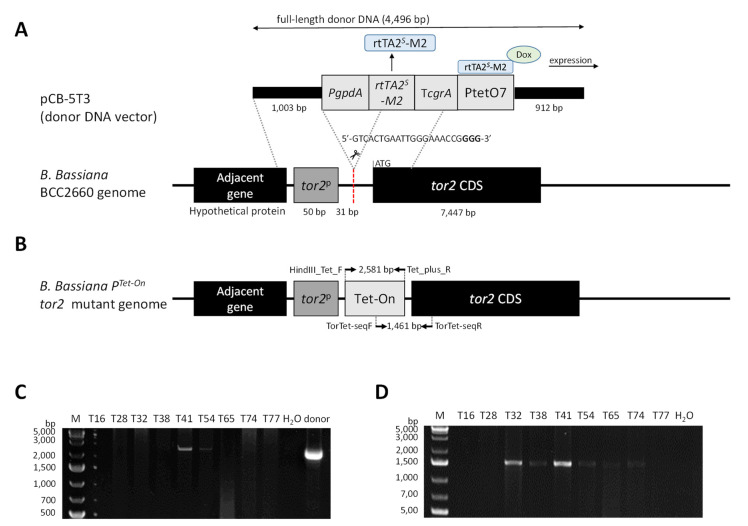
Genetic engineering of Tet-On upstream of *Beauveria bassiana tor2* coding sequence (CDS). (**A**) CRISPR/Cas9-mediated DSB targeted at 31 bp upstream of the start codon of *tor2*. Donor DNA vector pCB-5T3 consisted of the 2581 bp *Tet* sequence flanked by homology arms of 1003 bp and 912 bp corresponding to the promotor and CDS of *tor2*. (**B**) Schematic diagram of the *P*^Tet-On^ *tor2* mutant genome at the *tor2* locus. PCR verification of the integration site included amplification of *Tet* sequence by primer pair HindIII_Tet_F and Tet_plus_R and amplification of a *Tet-* and *tor2* CDS-inclusive sequence by primer pair TorTet-seqF and TorTet-seqR. (**C**) PCR products from nine transformants from the amplification using HindIII_Tet_F and Tet_plus_R. Lanes: M (GeneRuler 1kb plus DNA ladder); each of nine transformants; water (negative control) and donor (positive control). (**D**) PCR products from nine transformants from the amplification using TorTet-seqF and TorTet-seqR. Lanes: M, each of nine transformants and water (negative control).

**Figure 2 microorganisms-11-02129-f002:**
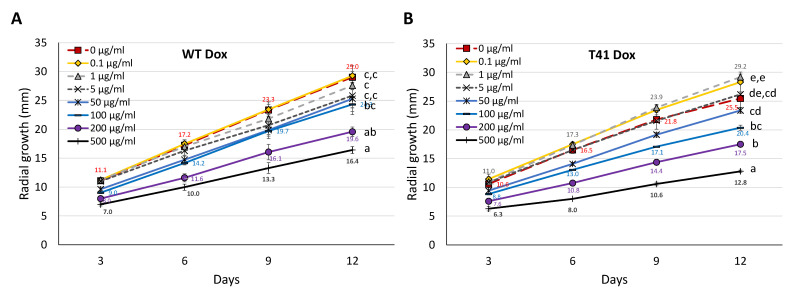
Vegetative growth of *Beauveria bassiana* WT and *P*^Tet-On^ *tor2* mutant T41 under various doxycycline (Dox) concentrations. Radial growths of WT (**A**) and T41 (**B**) in 12 days with Dox at 0 to 500 µg·mL^−1^. Data shown are mean ± SEM. Means with different letters are significantly different (Duncan’s, *p* < 0.05) (only radial growth data on day 12 are given).

**Figure 3 microorganisms-11-02129-f003:**
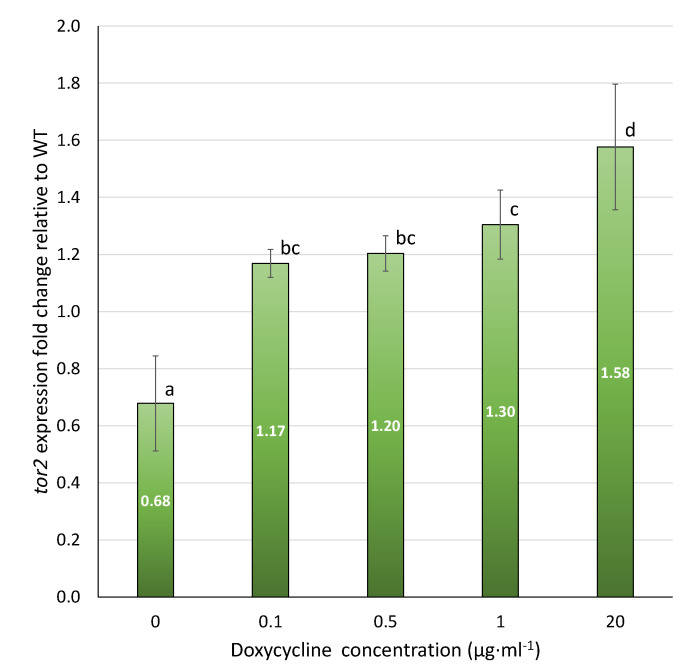
qPCR analysis of *tor2* expression in *P*^Tet-On^ *tor2* mutant T41, determined as fold changes relative to that of WT, under doxycycline concentrations from 0 to 20 µg·mL^−1^. Data shown are mean ± SEM. Means with different letters are significantly different (Duncan’s, *p* < 0.05).

**Figure 4 microorganisms-11-02129-f004:**
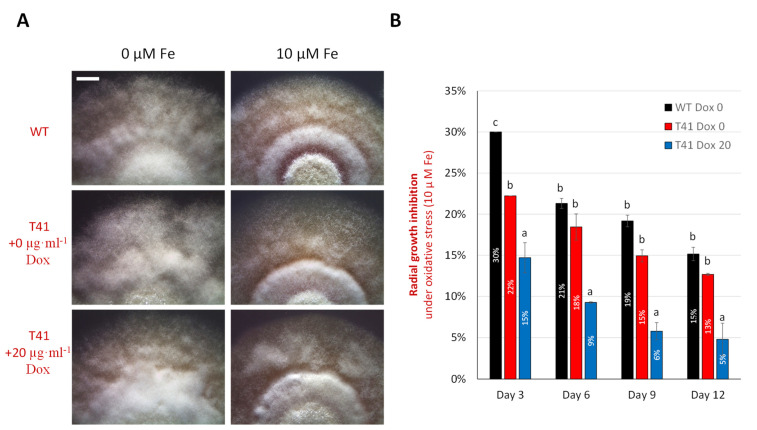
Oxidative stress tolerance of *Beauveria bassiana* WT and T41 under iron-replete condition (minimal medium agar supplemented with 10 µM Fe). T41 treatments were supplemented with 0 or 20 µg·mL^−1^ doxycycline (Dox). (**A**) Colony morphology of WT and T41 in various treatments on day 12. Bar, 2 mm. (**B**) Radial growth inhibition of WT and T41 on days 3, 6, 9 and 12 under the ferrous-ion-induced oxidative stress. Data shown are mean ± SEM. Means with different letters are significantly different (Duncan’s, *p* < 0.05).

**Figure 5 microorganisms-11-02129-f005:**
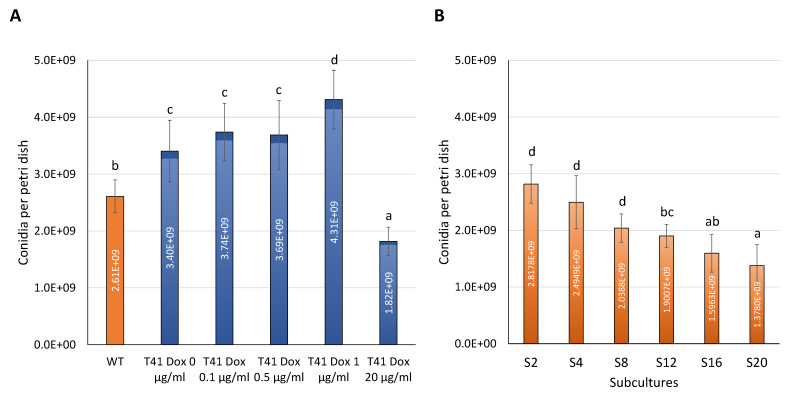
Conidiation of *Beauveria bassiana* WT and T41 with different doxycycline (Dox) concentrations. (**A**) Conidiation of T41 with 0 to 20 µg·mL^−1^ of Dox. (**B**) Conidiation of successive subcultures S2 to S20 in the WT. Data shown are mean ± SEM. Means with different letters are significantly different (Duncan’s, *p* < 0.05).

**Figure 6 microorganisms-11-02129-f006:**
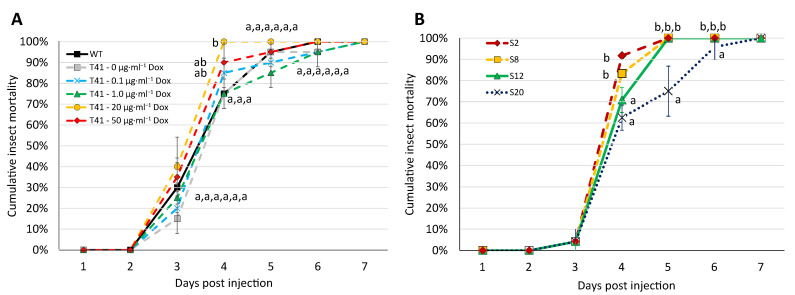
Virulence of *Beauveria bassiana* WT and *P*^Tet-On^ *tor2* mutant T41 against *Spodoptera exigua*. (**A**) Cumulative mortality of *S. exigua* after injection of T41 conidia under supplementation with 0 to 20 µg·mL^−1^ doxycycline (Dox). (**B**) Cumulative mortality of *S. exigua* after injection of WT conidia from each successive subculture of S2, S8, S12 and S20. Data shown are mean ± SEM. Means with different letters are significantly different (Duncan’s, *p* < 0.05).

**Table 1 microorganisms-11-02129-t001:** LT_50_ of *Spodoptera exigua* larvae after injection with *P*^Tet-On^ *tor2* mutant T41 in low Dox concentrations of 0, 0.1, 0.5, 1 and 20 µg·mL^−1^.

Treatment	LT_50_ (Days)
WT	4.19 ± 0.04 ^ab^
T41 Dox 0 µg·mL^−1^	4.28 ± 0.19 ^b^
T41 Dox 0.1 µg·mL^−1^	3.65 ± 0.11 ^d^
T41 Dox 1 µg·mL^−1^	3.74 ± 0.2 ^cd^
T41 Dox 20 µg·mL^−1^	3.08 ± 0.08 ^e^
T41 Dox 50 µg·mL^−1^	3.30 ± 0.14 ^de^

Data shown are mean ± SEM. Means with different letters are significantly different (Duncan’s *p* < 0.05).

**Table 2 microorganisms-11-02129-t002:** LT_50_ of *Spodoptera exigua* larvae after injection of different subcultures of WT *Beauveria bassiana*.

Subculture	LT_50_ (Days)
S2	3.55 ± 0.05 ^a^
S8	3.63 ± 0.05 ^ab^
S12	3.76 ± 0.02 ^b^
S20	4.16 ± 0.02 ^c^

Data shown are mean ± SEM. Means with different letters are significantly different (Duncan’s *p* < 0.05).

## Data Availability

The data presented in this study are available in the manuscript.

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
