# Peer review of "TOR Signaling Tightly Regulated Vegetative Growth, Conidiation, Oxidative Stress Tolerance and Entomopathogenicity in the Fungus Beauveria bassiana"

_microorganisms, 2023, doi:10.3390/microorganisms11092129_

Round 1
Reviewer 1 Report
the use of entomopathogenic fungi is highly desirable in agriculture. however, as pointed out by the authors, one of the main problems for the use of these biological control agents is their degeneration. Degeneration not only affects the virulence of isolates, but the shelf life of formulated products. In this context, I congratulate the authors for this work.
The introduction is clear and presents what motivated the research and its relevance. the material and methods topic is robust, with bioassays widely reported in the literature. the results are presented clearly and are in accordance with the bioassays performed. The discussion is adequate, based on the results, seeking to justify them and comparing them with the literature.
With the intention of contributing to the manuscript, I make some suggestions for consideration.
L38-39; 43-44: "biogical control agents" (BCA) can be macro-organisms or microorganisms, such as predators, parasitoids, fungi, bacteria and entomopathogenic viruses. The authors restrict BCA to entomopathogenic fungi, so I ask you to evaluate the use of the term.
L198: How were the Spodoptera exigua larvae obtained? I think that the insertion of the insect creation methodology is relevant.
Author Response
Reviewer I
The use of entomopathogenic fungi is highly desirable in agriculture. however, as pointed out by the authors, one of the main problems for the use of these biological control agents is their degeneration. Degeneration not only affects the virulence of isolates, but the shelf life of formulated products. In this context, I congratulate the authors for this work.
The introduction is clear and presents what motivated the research and its relevance. the material and methods topic is robust, with bioassays widely reported in the literature. the results are presented clearly and are in accordance with the bioassays performed. The discussion is adequate, based on the results, seeking to justify them and comparing them with the literature.
RESPONSE: We highly appreciate your kind comment and your recognition of our findings. The application of entomopathogenic fungi is key to sustainable and environmentally-friendly agriculture, which has exceptional economic importance in Thailand.
With the intention of contributing to the manuscript, I make some suggestions for consideration.
L38-39; 43-44: "biogical control agents" (BCA) can be macro-organisms or microorganisms, such as predators, parasitoids, fungi, bacteria and entomopathogenic viruses. The authors restrict BCA to entomopathogenic fungi, so I ask you to evaluate the use of the term.
RESPONSE: Thank you for your remark. we admit that the wordings in the paragraph - ‘…its widespread application as a biological control agent (BCA)’ - did not highlight sufficiently the wide variety of BCAs and could misguide readers. In our revision, we removed the term considering the introduction limit and the relevance to this article.
L198: How were the Spodoptera exigua larvae obtained? I think that the insertion of the insect creation methodology is relevant.
RESPONSE: We appreciate the comment. All the the Spodoptera exigua larvae were obtained from the the Viral Technology for Biocontrol Research Team (the facility at BIOTEC, Thailand). Consequently, we conducted the insect bioassay as described in the text.
Reviewer 2 Report
The authors in this article studied the inducible Tet-On promoter upstream of TOR kinase 2 gene tor2 using novel CRISPR/Cas9 technique in Beauveria bassiana fungus – the authors observed TOR kinase role in the vegetative growth, conidiation, pathogenicity and oxidative stress tolerance in the fungus. This ms is with in scope of journal, it is an interesting study and authors generated good data. The objectives of this study are clear, the experimental design is appropriate and the results support the conclusion. Overall, the article is good and can be accepted for its publication in microorganisms journal with following some minor changes:
P1, L13, provide clearly the Objective/s of study
P2, L63, provide complete genus name at the start of each sentence/paragraph throughout ms
P5, how many times this experiment was repeated (section 2.8)
P5, L206-210, was the date normalized and what was the model used to normalize the data, pl cite relevant citations for analysis/procedures and provide citation for statistical software/s both for SPSS and Probit analysis – how the significance letters were generated etc. – give detailed information in this section
P12, make necessary corrections in Author Contribution’s section
Table 1 and 2, give chi-square and fudicial limit values and other relevant values
It is suggested the sub-headings of Materials and Methods and Results should correspond each other for easy understanding of readers
Moderate editing in language is required for easy understanding of readers
Author Response
Reviewer II
The authors in this article studied the inducible Tet-On promoter upstream of TOR kinase 2 gene tor2 using novel CRISPR/Cas9 technique in Beauveria bassiana fungus – the authors observed TOR kinase role in the vegetative growth, conidiation, pathogenicity and oxidative stress tolerance in the fungus. This ms is with in scope of journal, it is an interesting study and authors generated good data. The objectives of this study are clear, the experimental design is appropriate and the results support the conclusion. Overall, the article is good and can be accepted for its publication in microorganisms journal with following some minor changes:
RESPONSE: We highly appreciate your comments. The TOR signaling pathway is an interesting network that is associated not only to growth but also secondary metabolism, stress response and potentially culture degeneration. We are glad to revise the manuscript accordingly.
P1, L13, provide clearly the Objective/s of study
RESPONSE: Thank you for your suggestion. The main objectives of the study is to determine the contribution of TOR kinase in growth and development in Beauveria bassiana BCC 2660, and to determine if the relevant contributions are dose dependent. We have included it in the last paragraph of ‘Introduction’ in our revised manuscript.
P2, L63, provide complete genus name at the start of each sentence/paragraph throughout ms
RESPONSE: Thank you for the suggestion. For the start of each sentence (also in figures) and of each paragraph, the full genus and species name ‘Beauveria bassiana’ has been given.
P5, how many times this experiment was repeated (section 2.8)
RESPONSE: We conducted the experiment twice. Thank you for your reminder. We have included it in the section 2.8.
P5, L206-210, was the date normalized and what was the model used to normalize the data, pl cite relevant citations for analysis/procedures and provide citation for statistical software/s both for SPSS and Probit analysis – how the significance letters were generated etc. – give detailed information in this section
RESPONSE: We highly appreciate your comment on the statistical analysis. For all our experiments, only the gene expression analysis (qPCR) part required the data normalization. The 2–∆∆Ct method method in the paragraph (section 2.5) included a normalization procedure. In addition, for the citations for SPSS and Probit, the IBM stated that “In many cases a formal reference or bibliographic citation is not necessary. For example, in popular APA style, SPSS is considered sufficiently known that only the version or release number is required.” Thus, we respectfully keep the statement as it was. Thank you very much. However, we have provided more details in this section (2.9).
P12, make necessary corrections in Author Contribution’s section
RESPONSE: We have made corresponding corrections in the Author Contributions section.
Table 1 and 2, give chi-square and fudicial limit values and other relevant values
RESPONSE: We appreciate the comment. The data in the two tables were normally distributed (as shown in the picture below; please see it in attached file), therefore we chose to do ANOVA analysis.
It is suggested the sub-headings of Materials and Methods and Results should correspond each other for easy understanding of readers
RESPONSE: Thank you for your suggestion. However, it would be quite challenging to realign the subheadings of Materials and Methods (M&M) and Results parts. As you may have noticed, the M&M’s sections 2.1-2.4 focused on the genetic engineering part led to the Results’ section 3.1. On the other hand, the Results’ sections 3.2 and 3.4 were from the M&M’s section 2.6. Therefore, we would like to indicate the subheadings in Results to reflect the main findings in our work. Again, we really appreciate the reviewer’s suggestion.
